# *Antrodia cinnamomea* Suppress Dengue Virus Infection through Enhancing the Secretion of Interferon-Alpha

**DOI:** 10.3390/plants11192631

**Published:** 2022-10-06

**Authors:** Yi-Ju Chen, Yu-Cian Tsao, Tzu-Chuan Ho, Irwin Puc, Chia-Chang Chen, Guey-Chuen Perng, Hsiu-Man Lien

**Affiliations:** 1Department of Microbiology & Immunology, College of Medicine, National Cheng Kung University, Tainan 70101, Taiwan; 2School of Medicine, College of Medicine, National Cheng Kung University, Tainan 701401, Taiwan; 3Department of Medical Imaging and Radiological Sciences, Kaohsiung Medical University, Kaohsiung 807, Taiwan; 4Institute of Basic Medical Sciences, College of Medicine, National Cheng Kung University, Tainan 701401, Taiwan; 5School of Management, Feng Chia University, Taichung 40724, Taiwan; 6Research Institute of Biotechnology, Hungkuang University, Taichung 43302, Taiwan

**Keywords:** dengue virus, *Antrodia cinnamomea*, megakaryocytes, interferon-alpha, inflammatory cytokines

## Abstract

Dengue caused by dengue virus (DENV) is a mosquito-borne disease. Dengue exhibits a wide range of symptoms, ranging from asymptomatic to flu-like illness, and a few symptomatic cases may develop into severe dengue, leading to death. However, there are no effective and safe therapeutics for DENV infections. We have previously reported that cytokine expression, especially inflammatory cytokines, was altered in patients with different severities of dengue. *Antrodia cinnamomea (A. cinnamomea)* is a precious and endemic medical mushroom in Taiwan. It contains unique chemical components and exhibits biological activities, including suppressing effects on inflammation and viral infection-related diseases. According to previous studies, megakaryocytes can support DENV infection, and the number of megakaryocytes is positively correlated with the viral load in the serum of acute dengue patients. In the study, we investigated the anti-DENV effects of two ethanolic extracts (ACEs 1–2) and three isolated compounds (ACEs 3–5) from *A. cinnamomea* on DENV infection in Meg-01 cells. Our results not only demonstrated that ACE-3 and ACE-4 significantly suppressed DENV infection, but also reduced interleukin (IL)-6 and IL-8 levels. Moreover, the level of the antiviral cytokine interferon (IFN)-α was also increased by ACE-3 and ACE-4 in Meg-01 cells after DENV infection. Here, we provide new insights into the potential use of *A. cinnamomea* extracts as therapeutic agents against DENV infection. However, the detailed mechanisms underlying these processes require further investigation.

## 1. Introduction

Dengue virus (DENV), a single-stranded RNA virus, belongs to the family *Flaviviridae* and genus *Flavivirus* [1,2]. There are four serotypes of DENV (DENV-1, DENV-2, DENV-3, and DENV-4) that can be distinguished by neutralization assay data [3]. Different serotypes may circulate in different areas and can be attributed to variations in severity. However, due to global warming, the increase in long-distance travel, population growth, and urbanization, cases of dengue have occurred in an increasing number of countries [4,5,6]. Dengue caused by DENV infection is one of the most important mosquito-borne diseases, with approximately four billion people living under the threat of dengue worldwide [1]. The clinical manifestations of dengue range from asymptomatic to flu-like, and some symptomatic cases develop severe dengue, including dengue hemorrhagic fever (DHF) and dengue shock syndrome (DSS) [7]. Severe dengue is life-threatening to patients, resulting in plasma leakage, organ failure, severe bleeding, cytokine storm, and even death [8,9]. In addition, we have previously reported that the expression of cytokines changes with different severities of dengue [10]. Many of the studies have also indicated that the levels of pro-inflammatory cytokines are elevated in severe dengue patients, such as interleukin (IL)-6, IL-8, tumor necrosis factor (TNF)-α, and IL-1β, which may result in cytokine storm and increase the severity of dengue in patients [11,12,13,14]. In contrast, type I interferon (IFN), including IFN-α and IFN-β, which are not only important human antiviral responses against pathogens but also the key factors linking the innate and adaptive immune systems, are suppressed by DENV and contribute to the production of DENV replication in host cells and evasion of antiviral immunity [15,16,17].

DENV has been around for many decades since 1943; however, there are no effective therapeutic agents against DENV infection. Currently, preventive medicine and supportive care are the main treatments administered to dengue patients [18]. Therefore, safe and efficient therapeutics against DENV infection are urgently required for the prevention and control of dengue. Dengvaxia, a live-attenuated dengue vaccine, is the first licensed dengue vaccine, but is recommended by the Food and Drug Administration (FDA) to be used only for children aged 9–16 years and who have been infected by DENV previously due to the vaccinated individuals who have not been infected by DENV, leading to an increased risk of developing severe dengue [19,20,21,22]. Although there are some candidates for antiviral agents for dengue that have entered clinical trials, none of them have been used to treat dengue patients [23].

*A**. cinnamomea* is a fungal parasite in the inner cavity of the endemic species *Cinnamomum kanehirae* (bull camphor tree) Hayata (Lauraceae). The host plant is a large evergreen broad-leaved tree that grows only in Taiwan and is distributed over broad-leaved forests at altitudes of 200–2000 m [24]. It is an edible Taiwanese mushroom that is regarded as a precious Chinese herbal medicine in Taiwan [25]. Many studies have reported that *A. cinnamomea* and its special components possess many pharmacological activities, including antihypertensive, neuroprotective, anticancer, and antimicrobial activities [26,27,28,29]. A recent study indicated that *A. cinnamomea* could inhibit angiotensin-converting enzyme 2 (ACE2), which supports severe acute respiratory syndrome coronavirus 2 (SARS-CoV-2) entry and disease onset [30]. Moreover, numerous studies have indicated that *A. cinnamomea* can inhibit the secretion of inflammatory cytokines, especially IL-6 or TNF-α, and suppress the surface antigens of the hepatitis B virus [31,32,33]. The ethanol extract of *A. cinnamomea* effectively suppressed hepatoma migration through downregulation of MAPK signaling, which exhibited potent DPPH radical- and superoxide dismutase (SOD)-like scavenging activities [34,35]. Furthermore, the anti-inflammatory compounds such as zhankuic acid C (ergostane-type triterpenoids), 4,7-dimethoxy-5-methyl-1,3-benzodioxole (benzenoids), and dehydrosulfurenic acid (lanostane-type triterpenoids) have been isolated and identified from the fruiting bodies and solid-state cultivated products of *A. cinnamomea* [24]. Active components isolated from *A. cinnamomea* have been demonstrated to be the major contributors to various medical benefits. Dehydroeburicoic acid can modulate glycolysis to prevent type 2 diabetes [36]. Zhankuic acids display an immunosuppressive effect on immune cells and could be used as a potential treatment for chronic inflammation [37,38]. 4,7-dimethoxy-5-methyl-1,3-benzodioxole induced cell apoptosis, resulting in decreased colon cancer cell growth and reduced pro-inflammatory cytokines by suppressing the nuclear factor-κβ (NF-κβ) signaling pathway [28,39,40]. Overall, *A. cinnamomea* is a potential source for novel strategies for developing new treatments for dengue.

Bone marrow (BM) suppression is a common phenomenon in dengue patients and leads to many hematological disorders such as leukopenia, thrombocytopenia, and atypical lymphocytes [41,42,43]. Furthermore, DENV directly infects and reduces the proliferative capacity of hematopoietic stem progenitor cells, which are BM progenitor cells, during DENV infection [44]. We have also demonstrated that megakaryocytes isolated from human BM are the dominant cells supporting DENV infection, and the number of megakaryocyte precursors was positively correlated with the viral load in the serum of patients with acute dengue [45]. Bone marrow-derived megakaryoblastic cells, Meg-01 cells, which are permissive for DENV infection, also show phenotypic properties similar to megakaryoblasts in the BM [46,47,48]. Hence, the hypothesis of this study was that the human megakaryocytic cell line Meg-01 as a cell model to explore the bioactivities of *A. cinnamomea* extracts against DENV infection.

## 2. Results

### 2.1. Cell Viability of the Five A. cinnamomea Extracts in Meg-01 Cells

Information on the five *A. cinnamomea* extracts is presented in Table 1. The WST-1 cell proliferation assay kit was used to assess the viability of Meg-01 cells after treatment with *A. cinnamomea* extracts. The results showed that each *A. cinnamomea* extract caused concentration-dependent cytotoxicity in Meg-01 cells (Figure 1). The values of CC_10_ and CC_50_ values were calculated to select an appropriate concentration of the *A. cinnamomea* extract for the following antiviral activity test (Table 1). Of the five *A. cinnamomea* samples, ACE-4 demonstrated the lowest cytotoxicity, whereas ACE-1, ACE-2 and ACE-5 induced higher levels of cell death in Meg-01 cells. The values of CC_10_ were between 3.84 and 5.68 μg/mL, and the values of CC_50_ were between 35.67 and 55.96 μg/mL.

### 2.2. Inhibitory Effects of the A. cinnamomea Extracts against DENV

Meg-01 cells were treated with different *A. cinnamomea* extracts after DENV infection, and the viral titers of DENV were evaluated using a plaque assay (Figure 2). The concentrations of ACE-1–ACE-5 were used in viral titer, which are 3.84, 5.67, 91.68, 176.51, 5.68 μg/mL, respectively. The replication curve for DENV in Meg-01 cells was calculated. The viral titer peak was observed on day 5 post-infection and remained steady until day 7 (Figure 2a). The antiviral effects of *A. cinnamomea* extracts on DENV were determined by performing a plaque assay with the supernatants collected on different days post-infection. From the results, it was observed that the viral titers of DENV decreased in both ACE-3 and ACE-4 treatments during DENV infection as of day 2. Moreover, a significant reduction in DENV resulting from ACE-3 and ACE-4 treatments was also observed at days 5 and 7 post-infection (Figure 2b). Although there was a visible decrease in the viral titer of DENV at 24 h after DENV infection in ACE-1, ACE-2, and ACE-3 treatments, the difference was not statistically significant.

### 2.3. Anti-Inflammatory Effects of the A. cinnamomea Extracts

Disorder in cytokine expression is one of the clinical manifestations of dengue. To date, much of the evidence suggests that this dysfunction is correlated with disease severity. Therefore, we evaluated the ability of *A. cinnamomea* extract to regulate cytokine expression. The supernatants of *A. cinnamomea* extract-treated Meg-01 cells after DENV infection were collected to analyze the secretion patterns of inflammatory cytokines, such as IL-6, IL-8, and TNF-α. The results indicated that the expression of IL-6 and IL-8 in control group which was Meg-01 cells without *A. cinnamomea* extract treatment was associated with the dynamics of dengue viremia. Notably, after treatment with the *A. cinnamomea* extracts, ACE-3 and ACE-4, the expression of both IL-6 and IL-8 was significantly decreased after DENV infection (Figure 3). As a result, ACE-3 and ACE-4 efficiently reduced the inflammatory cytokines IL-6 and IL-8, in Meg-01 cells against DENV infection.

### 2.4. Antiviral Activities of the A. cinnamomea Extracts

As mentioned previously, DENV can suppress the antiviral activities of the host, especially impeding the signaling response of type I IFN, to promote viral replication. Therefore, we analyzed the production of type I IFN, IFN-α, and IFN-β. Increased levels of IFN-α were observed in Meg-01 cells treated with *A. cinnamomea* extracts after DENV infection when compared to the control, which were only infected with DENV (Figure 4a). Additionally, the level of IFN-β fluctuated and had no relationship with the viral titers or *A. cinnamomea* extract treatment (Figure 4b). Taken together, *A. cinnamomea* extracts, ACE-3, and ACE-4, increased the secretion of IFN-α to suppress DENV replication.

## 3. Discussion

In recent years, some investigational drugs against DENV infection entered into the clinical development phase but failed during or after the clinical trials [23]. Although there are many standard operating procedures that may mitigate the prevalence and occurrence of dengue fever, the healthcare costs involved are high. Therefore, there is an increasing requirement for dengue therapeutics and an impetus for antiviral drug and vaccine development in this realm. In the past century, natural components such as plants and microorganisms, which have biological activities, have emerged as one of the major sources for developing new pharmaceuticals [49,50,51].

Herbal medicine refers to botanicals that have been modified and successfully applied to treat many diseases [52,53,54]. According to the WHO, herbal medicine is widely used for health care, disease prevention, and treatment in many countries [55]. *A. cinnamomea* has been revealed to have diverse activities, and the active components in *A. cinnamomea* are important ingredients used in the production of health food products [56,57].

In this study, we focused on the antiviral and anti-inflammatory effects of *A. cinnamomea* in DENV infection. We found that ACE-3 and ACE-4 have potential therapeutic effects against DENV by enhancing the antiviral cytokine expression and suppressing the inflammatory cytokine secretion to inhibit DENV replication in Meg-01 cells. These protective effects of *A. cinnamomea* were examined by pre-treating Meg-01 cells with *A. cinnamomea* extract before DENV infection. Moreover, it was found that the activity of type I interferons increased early during DENV infection when Meg-01 cells were treated with ACE-3 or ACE-4, in comparison to that without treatment. This result implied that the components of *A. cinnamomea* appeared to repair IFN damage. DENV life cycle can be divided into three stages: entry, replication, and release. To investigate the potent inhibitory mechanism of *A. cinnamomea* extract against DENV infection, Meg-01 cells were treated with *A. cinnamomea* extract and DENV at various incubation periods and temperatures in a previous study to find out the target molecule or the signaling pathway altered by the *A. cinnamomea* extract [58].

ACE-3 (Zhankuic acid C), one of the major components ofthe fruiting bodies of *A. cinnamomea*, has been isolated from the extract by using HPLC in the study. It has been reported to have anti-inflammatory effects in neutrophils and dendritic cells, thereby treating chronic inflammation and auto immune diseases [37]. ACE-4 (4,7-dimethoxy-5-methyl-1,3-benzodioxole), which was also isolated from the fruiting bodies of *A. cinnamomea* and found to display a remarkable inhibitory effect in LPS-induced inflammation [39]. In the past, the fruiting bodies of *A. cinnamomea* have been used for the prevention or treatment of numerous diseases including liver diseases, food and drug intoxication, diarrhea, abdominal pain, hypertension, itchy skin, and tumorigenic disease [24]. However, the presence of any anti-viral effect of *A. cinnamomea* extract has barely been explored. Likewise, the relevance of *A. cinnamomea* components’ function to DENV has not been evaluated to date. The discovery of IFN damage repair by *A. cinnamomea* components was a momentous breakthrough in the development of anti-DENV drugs.

## 4. Materials and Methods

### 4.1. Cells and Viruses

Baby hamster kidney fibroblast cells (BHK-21) (ATCC, Manassas, VT, USA) and African green monkey kidney epithelial cells (Vero) (ATCC, Manassas, VT, USA) were cultured and maintained in Dulbecco’s modified Eagle´s medium (DMEM) (Cytiva, Chicago, IL, USA) supplemented with 5% heat-inactivated fetal bovine serum (FBS) (Gibco, Waltham, MA, USA). Philadelphia chromosome-positive chronic myelogenous leukemia bone marrow cells (Meg-01) (ATCC, Manassas, VT, USA) were cultured and maintained in RPMI-1640 medium (RPMI) (Cytiva, Chicago, IL, USA) supplemented with 10% heat-inactivated FBS. All the cells were grown at 37 °C with 5% CO_2_. Subculturing procedures were performed according to the guidelines provided by ATCC. Dengue virus serotype 2 (DENV, 16681 strain) was used in these experiments [59]. DENV was propagated in Vero cells, and the viruses were titrated by plaque assay and used for the following experiments.

### 4.2. A. cinnamomea Extracts Preparation

Fruiting bodies of *A. cinnamomea* were collected from Yuli, Hualien County (23°24′32.5″ N, 121°24′49.1″ E), Taiwan, in December 2007. The extracts were prepared as follows: crude extracts of *A. cinnamomea*: 20 g of ground *A. cinnamomea* (passed through a 30 mesh screen) was mixed in 50% ethanol solution and 95% ethanol solution and stirred at 125 rpm for 2 days at room temperature. The extracts were then filtered using medium-grade filter papers and concentrated under reduced pressure at 35 °C to give a dark brown syrup.

ACE-3, ACE-4, and ACE-5 isolation: The dried and pulverized *A. cinnamomea* fruiting body samples were extracted thrice with 95% ethanol at a ratio of 1:10 (*w*/*v*) for 2 days at room temperature under constant stirring at 125 rpm. After two days, the ethanol solution was filtered and concentrated to dryness in vacuo. The crude extract was suspended in deionized water and successively partitioned with ethyl acetate (EtOAc). The concentrated EtOAc layer was chromatographed on a silica gel column by elution with an n-hexane/EtOAc gradient with increasing polarity, yielding five fractions. The fractions were then analyzed using high-performance liquid chromatography to obtain ACE-3, ACE-4 and ACE-5, respectively. The structures of the compounds (ACE-3, ACE-4, and ACE-5) were determined by ^1^H and ^13^C nuclear magnetic resonance spectroscopy and by comparison of the spectral data with published values [60,61,62,63]. A summary of the current protocol is shown in Figure 5.

### 4.3. Cell Viability Test

Cell viability was measured using Cell Proliferation Reagent WST-1 (Roche, Basel, BS, Switzerland). In brief, Meg-01 cells were seeded at a density of 1000 cells per well in 100 μL culture medium containing gradient concentrations of *A. cinnamomea* extracts in 96-well plates. Cells cultured in a medium without *A. cinnamomea* extract were used as control cells. The plates were incubated at 37 °C in an incubator with 5% CO_2_ for 7 days. After incubation, 10 μL of WST-1 assay reagent was added to each well and incubated for another 2 h. Absorbance at 440 nm was determined using theenzyme-linked immunosorbent assay (ELISA) reader (Multiskan SkyHigh Microplate Spectrophotometer, Thermo Fisher Scientific, Waltham, MA, USA). All experiments were performed in duplicate three times. According previous studies described [64,65,66], the values of CC_10_ and CC_50_ (cytotoxicity concentration that causes a 10% and 50% reduction in cell numbers compared to the control) were calculated a 4-parameter logistic (4PL) model and were regarded as the working concentration of the *A. cinnamomea* extracts for the virus inhibition assay.

### 4.4. Virus Inhibition Assay

Meg-01 cells were transferred into the flow tube at a density of 2 × 10^5^ per tube in 1 mL culture, followed by DENV infection at a multiplicity of infection (MOI) of 1. The tubes were incubated at 37 °C incubator with 5% CO_2_. After incubation, the cells were washed with fresh culture medium and resuspended in a culture medium containing different *A. cinnamomea* extracts. Cells resuspended in the culture medium without *A. cinnamomea* extract were used as control cells. The supernatants were collected 1, 2, 3, 5, and 7 days post infection and stored at −80 °C for subsequent titration by plaque assay.

### 4.5. Plaque Assay

BHK-21 cells were seeded at a density of 5 × 10^5^ cells/well in 1 mL of culture medium in 6-well plates. Serial dilution (10^−1^–10^−6^) was performed by adding 100 μL supernatants or the virus to 900 μL DMEM supplemented with 2% FBS to make a 1 to 10 dilution. After the cells were attached, the medium was removed, and 400 μL of the serial dilutions was added to each well. The plates were incubated at 37 °C and 5% CO_2_ for 2 h and shaken every 15 min to prevent the cells from drying. After the incubation, supernatants were removed, and 2 mL 1% methyl cellulose (Sigma-Aldrich, St. Louis, MO, USA) medium containing with 1% L-glutamine (Cytiva, Chicago, IL, USA), 1%sodium pyruvate (Cytiva, Chicago, IL, USA), 1%sodium bicarbonate (Cytiva, Chicago, IL, USA), 20 mL FBS, 1% HEPES (Cytiva, Chicago, IL, USA) was added to each well. Seven days after incubation, the cells were stained with crystal violet and the plaques were counted to determine the viral titers. The viral titers were calculated using the following formula:$$\mathrm{Viral}\;\mathrm{titer}\left(\frac{\mathrm{PFU}}{\mathrm{mL}}\right)\;=\;\frac{\mathrm{Average}\;\mathrm{number}\;\mathrm{of}\;\mathrm{the}\;\mathrm{plaques}\;x\;\mathrm{the}\;\mathrm{volume}\;\mathrm{of}\;\mathrm{serial}\;\mathrm{dilutions}\;\mathrm{per}\;\mathrm{well}}{\left(\mathrm{Dilution}\;\mathrm{factor}\right)}$$

### 4.6. Detection of Inflammatory Cytokines

The supernatants collected from the virus inhibition assay were analyzed using enzyme-linked immunosorbent assay (ELISA) to detect the expression of cytokines, including IL-6 (Invitrogen, Waltham, MA, USA), IL-8 (Invitrogen, Waltham, MA, USA), TNF-α (Invitrogen, Waltham, MA, USA), IFN-α (Invitrogen, Waltham, MA, USA), and IFN-β (Invitrogen, Waltham, MA, USA). The assay was performed according to the manufacturer’s instructions.

### 4.7. Statistic

The results are presented as the mean standard error of the mean (mean ± SEM). Statistical analyses were performed using analysis of variance (ANOVA) or the Mann–Whitney test, and the significant differences were marked as * (*p* < 0.05), ** (*p* < 0.01), *** (*p* < 0.001), and **** (*p* < 0.0001), respectively. All results and statistical calculations were performed using Microsoft Excel 2019 and GraphPad Prism version 9.0.

## 5. Conclusions

We can conclude that *A. cinnamomea* may be a potential source for developing antiviral drugs or therapeutics for dengue fever by decreasing the inflammatory response and enhancing antiviral cytokine secretion. These findings provide new insights into the possible use of *A. cinnamomea* extract as a therapeutic agent against DENV infections.

## Figures and Tables

**Figure 1 plants-11-02631-f001:**
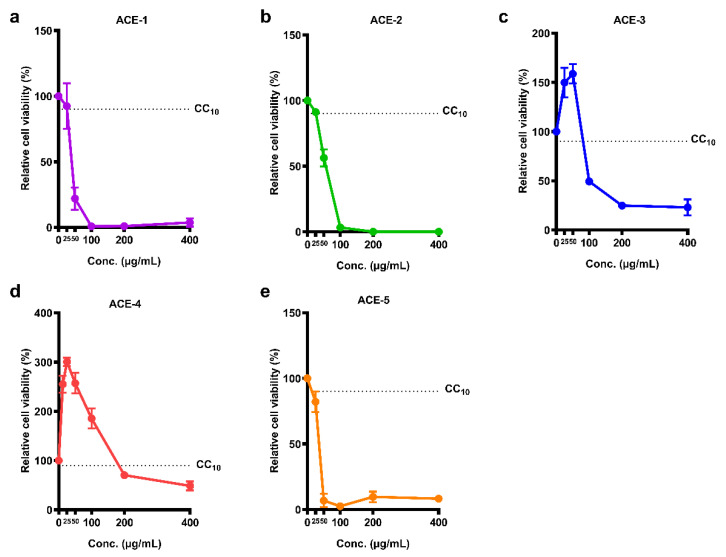
Cell viability of the five *A. cinnamomea* extracts in Meg-01 cells. Meg-01 cells were cultured with different concentrations of *A. cinnamomea* extracts, (**a**) ACE-1, (**b**) ACE-2, (**c**) ACE-3, (**d**) ACE-4, and (**e**) ACE-5, for 24 h followed by reacting with WST-1 reagent. All the values represent as mean ± SEM of three experiments. The dotted line represents the value of CC_10_.

**Figure 2 plants-11-02631-f002:**
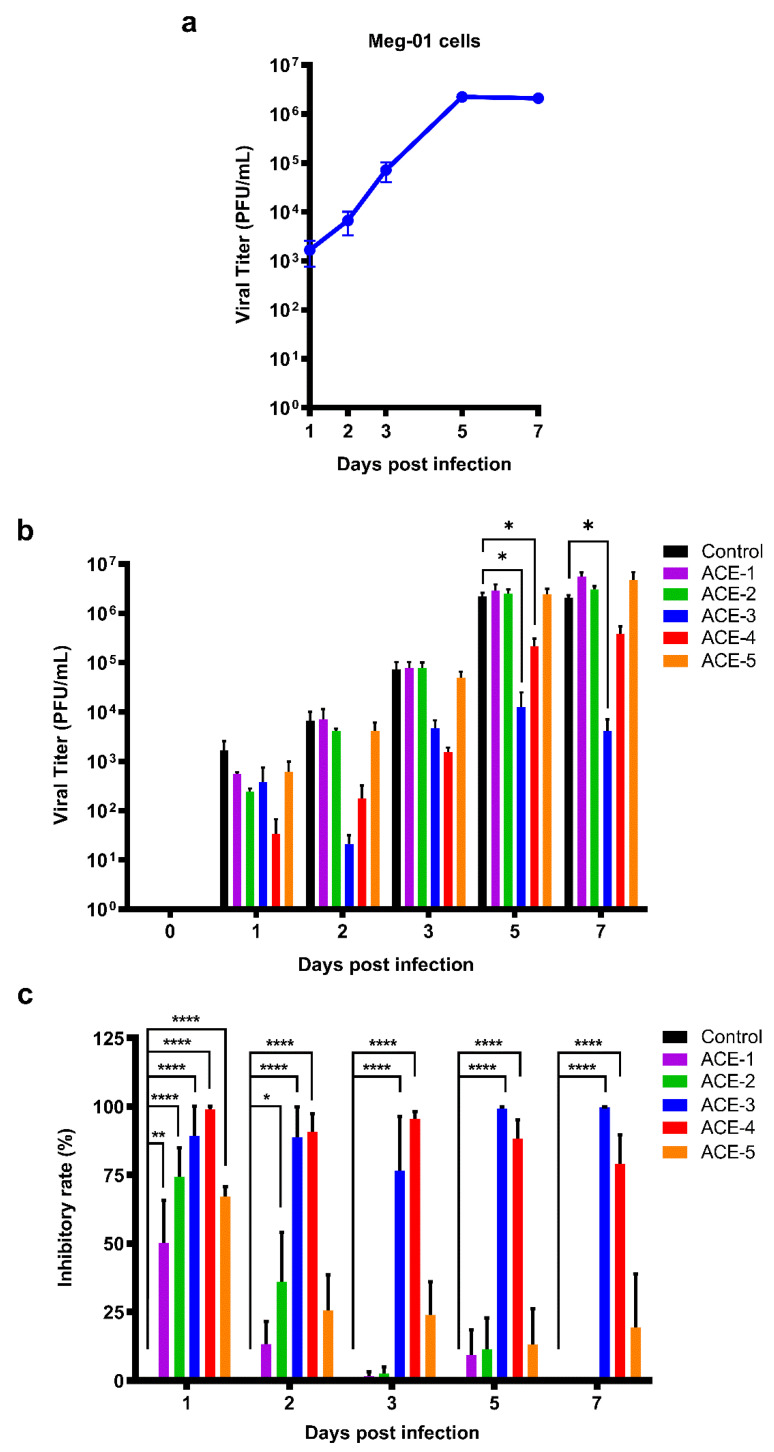
Both ACE-3 and ACE-4 could significantly inhibit DENV replication. (**a**) Replication curve of DENV in Meg-01 cells, and the peak of the titer was showed at day 5 post infection. (**b**) The viral titers of DENV in the *A. cinnamomea* extracts-treated Meg-01 cells. (**c**) The inhibitory rate of the *A. cinnamomea* extracts against DENV infection in Meg-01 cells. All the values represent as mean ± SEM of three experiments. *p* values were calculated using two-way ANOVA, statistically significant difference * *p* < 0.05, ** *p* < 0.01, **** *p* < 0.0001.

**Figure 3 plants-11-02631-f003:**
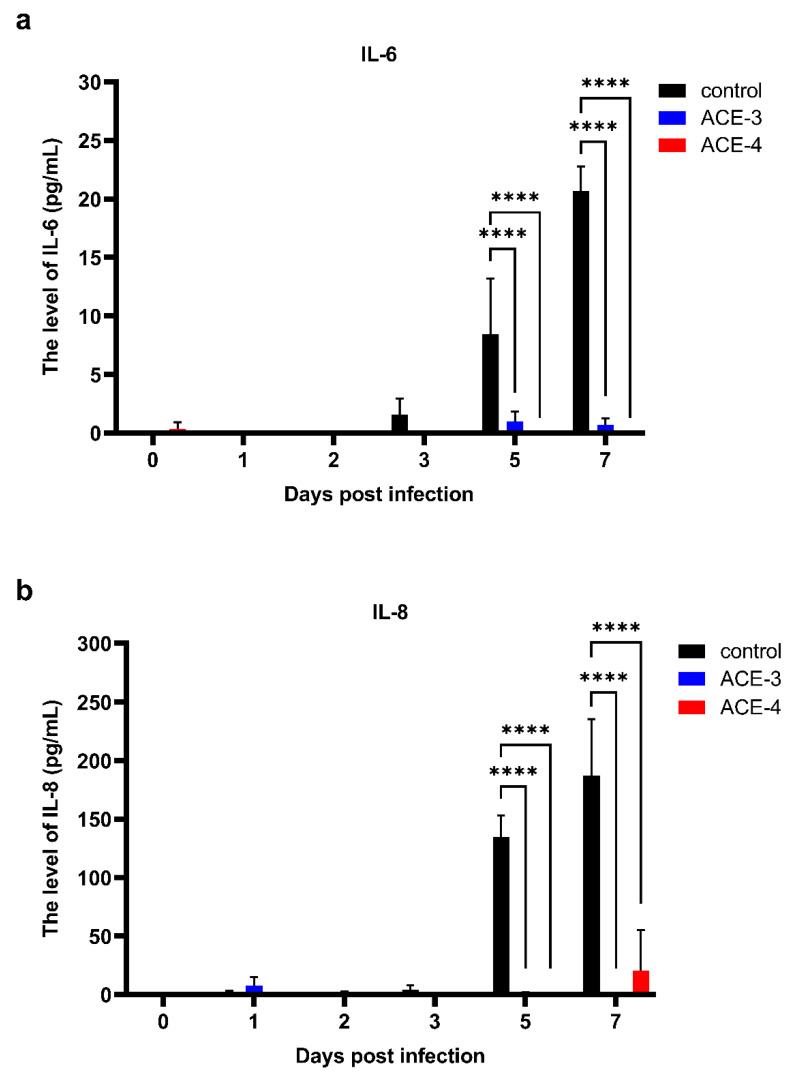
ACE-3 and ACE-4 suppressed the expression of the IL-6 and IL-8 after DENV infection. The levels of (**a**) IL-6 and (**b**) IL-8, in Meg-01 cells treated with the *A. cinnamomea* extracts after DENV infection. All the values represent as mean ± SEM of three experiments. *p* values were calculated using two-way ANOVA, *p* < 0.0001 **** compared to the control.

**Figure 4 plants-11-02631-f004:**
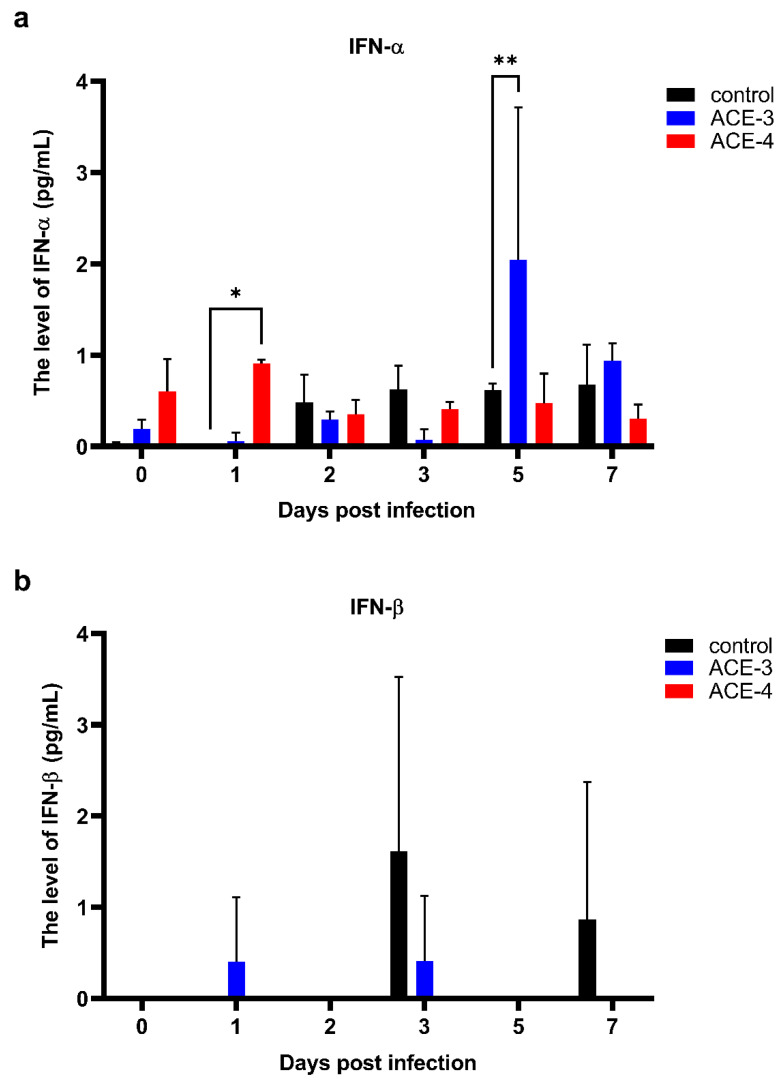
ACE-3 and ACE-4 may inhibit DENV replication by enhancing the secretion of IFN-α. The levels of antiviral cytokines, (**a**) IFN-α and (**b**) IFN-β, in Meg-01 cells treated with *A. cinnamomea* extracts after DENV infection. All the values represent as mean ± SEM of three experiments. *p* values were calculated using two-way ANOVA, statistically significant difference * *p* < 0.05, ** *p* < 0.01 compared to the control.

**Figure 5 plants-11-02631-f005:**
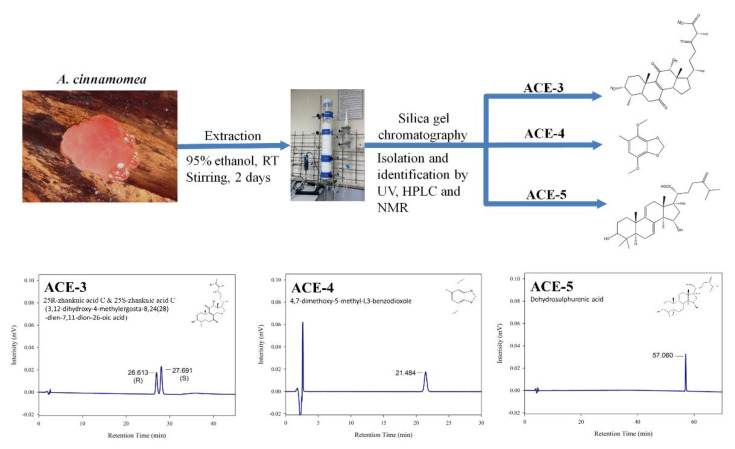
Methods for isolation and identification ofactive compounds from *A. cinnamomea*.

**Table 1 plants-11-02631-t001:** The inhibition effects of prepared two ethanolic extracts and isolated three compounds from *A. cinnamomea* on Meg-01 cells.

Groups	*A. cinnamomea* Extracts	Molecular Weight (g/mol)	CC_10_ (μg/mL)	CC_50_ (μg/mL)
ACE-1	Crude Extracts I (95% ethanol)	-	3.84	40.01
ACE-2	Crude Extracts II (50% ethanol)	-	5.67	55.96
ACE-3	Zhankuic acid C	486.64	91.68	99.69
ACE-4	4,7-dimethoxy-5-methyl-1,3-benzodioxole	196.07	176.51	387.59
ACE-5	DehydrosulphurenicAcid	486.00	5.68	35.67

## Data Availability

Not applicable.

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
