# Peer review of "Antrodia cinnamomea Suppress Dengue Virus Infection through Enhancing the Secretion of Interferon-Alpha"

_plants, 2022, doi:10.3390/plants11192631_

Round 1
Reviewer 1 Report
Review report
Dear editor and authors,
The interesting manuscript titled as “Antrodia cinnamomea suppress dengue virus infection through 2 enhancing the secretion of interferon-alpha” is a novel and significant contribution to science. In this work, Chen and coworkers evaluated the effects of five A. cinnamomea extracts (ACE-1 to ACE-5) against DENV 27 infection in Meg-01 cells. Our results not only demonstrated that ACE-3 and ACE-4 significantly 28 suppress DENV infection but also reduced interleukin (IL)-6 and IL-8. Moreover, the level of anti-29 viral cytokine, interferon (IFN)-α, was also increased by ACE-3 and ACE-4 in Meg-01 cells after 30 DENV infection. The manuscript is simple, well-organized and easy to read. The theory is also easy to understand and I think it will be read by many researchers. However, before the paper can be considered for publication/acceptance, it is necessary for authors to undertake minor revisions in accordance with the comments of as suggested by me. I hope that this article will appeal wide readership attention and will accelerate progress in this specific field. However, I do find some minor mistakes/shortcomings in the article that the author need to correct before publication. The manuscript is well written but I would recommend revision and improvement of certain sentences. I found this article insightful, applied and quite informative and recommend it for publication with minor revision.
ü The abstract is well-structured, well written and has excellently discussed and concluded the different aspects. However, it would be highly appreciated if authors incorporate sentence about the future perspectives of the research work conducted from present study. There is no future perspective statement in the abstract section. Provide a statement that what next can be done after these ethnobotanical surveys.
§ Why Portulacaria afra Jacq. and Setaria glauca L. has been chosen.
§ Please enter the geographic coordinates of Antrodia cinnamomea.
§ Please indicate clearly
ü Purpose
ü Research hypothesis
§ The introduction section is quite informative and to the point. Authors have comprehensively discussed the notion behind present research work. Authors are advised once again to double check grammar, sentence structure etc, if there any deficiency, fix them accordingly.
v Ther is no discussion of phytochemistry and mycochemistry. Provide 1-2 lines to support the theme of work performed. Read, Follow and cite the article with links provided below.
ü https://www.sciencedirect.com/science/article/abs/pii/S0022286022003696
ü https://www.nature.com/articles/s41598-021-99839-z
ü https://pubmed.ncbi.nlm.nih.gov/33332709/
§ There is no discussion on the overall medicinal plants flora and mushroom species of the said geographical area. Provide few words and discuss about the dominant medicinal plants in nearby areas.
§ I found few grammatical and spelling mistakes that can easily be corrected during proof reading.
§ The authors are advised to manage all the references in the same format.
§ Each abbreviation must be explained
§ Please add photos of the analyzed Antrodia cinnamomea.
§ What was ecological distribution and phytogeography, utilization status and threats to the surveyed medicinal mushroom used? Do you have some recommendations regarding their conservation aspects?. I think you must add 1-2 lines.
§ Great efforts have been made to write Materials and methods section. All protocols have been explained with full details which is quite fascinating for future researchers to easily pursue their experimental research data collected from field.
§ Results and discussion section is well written, all results have been rightly discussed with relavent data from literature. Add some new 2020-2021 data from literature if available.
§ The conclusion section is comprehensive and well-articulated. Remove results and introductory statements from conclusion section after carefully reviewing manuscript.
§ There is no uniformity in references. Cross check all the references and strictly follow plants MDPI author guidelines.
Author Response
Dear Reviewer :
Please refer to the attached file.

Reviewer 2 Report
The authors designed their study to investigate antiviral activity of Antrodia cinnamomea extracts against dengue virus by the secretion of interferon-alpha. However, the authors need to improve their manuscript before it consider for publishing. Below some of my comments for the authors to improve their study:
1. To make the manuscript sound scientific I suggest the ours don't use terms such as "We or Our" in their writing.
2. authors calculated CC10 value of their extract but it is important to calculate CC50 value as well. The authors also didn't mention what was the incubation time they tested cytotoxicity of their extracts
3. It will be more appropriate if the author we more descriptive in their results rather that refer everything to the figures and tables. it is quite challenging to understanding the results because their authors weren't descriptive enough in their results section.
4. Regarding the antiviral activity the authors need to calculate the IC50/EC50 of their extract. And base on the IC50 and CC50 they need to calculate their extract Selectivity Index (SI value). I believe the author only tested 1 concentration of each extract. it would be interesting if the authors choose the most effective extracts and perform antiviral test using few concentration of their extract to observe the pattern of viral reduction and calculate IC50 and SI value
5. In Discussion Section line 173-178 has repeated in introduction section.
6. Discussion section its more look like a literature review and not much of manuscript data discussion. The authors need to rewrite their discussion in the pattern of discussing their study finding.
Author Response
Dear Reviewer:
Please refer to the attached file.

Reviewer 3 Report
In manuscript plants-1915585 the authors have reported the antiviral activity of extracts and isolated compounds from the mushroom Antrodia cinnamomea against dengue virus. Both zhankuic acid C and 4,7-dimethoxy-5-methyl-1,3-benzodioxole were found to suppress viral infection. In addition, the compounds reduced pro-inflammatory cytokines interleukin-6 and interleukin-8 levels as well as increase the level of antiviral interferon-α. The manuscript is suitable for publication in Plants.
Some minor suggestions for the authors to consider:
1. The Introduction is a long, run-on paragraph that could be broken down to five paragraphs.
a. New paragraph starting with “DENV has been around…” (line 58)
b. New paragraph starting with “Atrodia cinnamomea…” (line 68)
c. New paragraph starting with “Bone marrow…” (line 85)
2. The first paragraph of the Discussion section could also be split. Consider starting a new paragraph with “Herbal medicine known as…” (line 186)
3. The penultimate version of the manuscript would benefit by a proofreading by an English-speaking technical editor.
Author Response

(The authors gave the same response as above.)

Round 2
Reviewer 2 Report
The authors of Manuscript entitle "Antrodia cinnamomea suppress dengue virus infection through 2 enhancing the secretion of interferon-alpha" have revised the manuscript satisfactory. However there are few things are not clear in the manuscript.
1. The concentration of extracts the authors used to treat dengue virus
2. The IC50 value and Selectivity Index value of extract against dengue virus
Author Response
Dear reviewer.
Please refer to the attached file.
